# Zoonotic Sporotrichosis outbreak: Emerging public health threat in the Amazon State, Brazil

Viviany Araujo Mesquita[1], Sinesio Talhari[1,2], André Luiz Leturiondo[1,2], Guilherme Caldas de Souza[1], Euzenio Moreira de Brito[1], Suanni Lemos de Andrade[1,3], Débora Cristina de Lima Fernandes[2], Maria Zeli Moreira Frota[4], Rossilene Conceição da Silva Cruz[1,2], Juliana de Andrade Rebouças Guimarães[1,5], Helio Amante Miot[6], Carolina Talhari[1,7] *, Valderiza Lourenço Pedrosa[1,2]

1 Programa de Pós-Graduação em Ciências Aplicadas à Dermatologia, Universidade do Estado do Amazonas, Manaus, Amazonas, Brazil, 2 Fundação Hospitalar Alfredo da Matta de Dermatologia, Manaus, Amazonas, Brazil, 3 Departament of Pathology—Universidade do Estado do Amazonas, Manaus, Amazonas, Brazil, 4 Universidade Federal do Amazonas, Manaus, Amazonas, Brazil, 5 Departament of Internal Medicine—Universidade do Estado do Amazonas, Manaus, Amazonas, Brazil, 6 Departamento de Dermatologia da FMB-UNESP, Botucatu, São Paulo, Brazil, 7 Departamento de Dermatologia da Universidade do Estado do Amazonas, Manaus, Amazonas, Brazil

* carolinatalhari@gmail.com

**Data Availability Statement:** Data is available from public records at Fundação de Vigilância em Saúde do Amazonas homepage (https://www.fvs.am.gov. br).

## Abstract

### Background

Sporotrichosis is the most common subcutaneous mycosis caused by *Sporothrix spp*. Traditionally, it is transmitted through injuries involving plant debris. However, over the past few decades, there has been an epidemic increase in human cases resulting from contact with infected animals, particularly cats, in various regions of Brazil. In this report, we report a notable increase in both human and animal cases within the Brazilian Amazon state.

### Methodology/Principal findings

An ecological study was conducted by analyzing official records of human and animal sporotrichosis diagnosed in the state of Amazon from 2020 to 2023. Data including patient demographics, clinical manifestations, mycological examination results, and species identification through PCR confirmation were evaluated. During this period, a total of 950 human cases and 2,823 animal cases of sporotrichosis were reported at an exponential rate, since no human cases were registered in 2020. The spatial and temporal dispersion of human sporotrichosis followed that of animal cases, moving from downtown areas to the periphery. Contact with infected animals was reported in 77.7% of cases, with cats being the most commonly implicated (73.5%). Only 66.7% of individuals underwent mycological examination. Among the positive cultures for *Sporothrix spp*., 65.4% were identified as *S. brasiliensis*. All patients were treated with systemic antifungals.

### Conclusions/Significance

This study highlights a rising incidence of sporotrichosis among animals and humans in the Brazilian Amazon region over the past four years, with *S. brasiliensis* being the predominant agent. Collaborative efforts involving healthcare professionals, veterinarians, and public

**Funding:** This study was supported by FAPEAM (Fundação de Amparo à Pesquisa do Estado do Amazonas, Brazil) through "Programa de Apoio à Formação em Ciências Dermatológicas – PRODERM-RH (grant #010/2023). HAM is a PVN-II Research Fellow from FAPEAM. The funders had no role in study design, data collection and analysis, decision to publish, or preparation of the manuscript.

**Competing interests:** The authors have declared that no competing interests exist.

health authorities are crucial to implement effective control measures, educate populations at risk, and promote responsible guidance for pet guardians. These measures are essential to mitigate the burden of epidemic sporotrichosis in Brazil.

## Author summary

Sporotrichosis is the leading subcutaneous mycosis worldwide. In the last decades, Brazil has faced an epidemic of zoonotic cases. The Brazilian Amazon region had no human cases reported in 2020, nevertheless, a striking rise in both human and animal cases was observed from 2020 to 2023, totalizing 950 human and 2,823 animal sporotrichosis. The majority of cases were reported in Manaus, the largest city in the Amazon, but cases were also documented in other populous municipalities. The study emphasizes the correlation between animal and human cases, as well as the spatial and temporal progression of the disease, moving from downtown areas to the periphery. Cats were identified as the primary reservoirs, with contact with infected animals being a significant risk factor for transmission. Urban areas, particularly domestic environments, were identified as common sites of infection. The predominant species identified was *Sporothrix brasiliensis*, which exhibits distinct characteristics favoring zoonotic transmission. This study emphasizes the urgent need for collaborative efforts among healthcare professionals, veterinarians, and public health authorities to implement effective control measures and mitigate the impact of the epidemic sporotrichosis in Brazil.

## Introduction

Sporotrichosis, the most common human subcutaneous fungal infection worldwide, is a neglected disease caused by various fungi of the genus *Sporothrix*. It primarily affects the skin and subcutis but can also involve the lymph nodes, bones, joints, and internal organs in rare cases, especially among immunosuppressed [1,2].

*Sporothrix spp*. (1) are thermal dimorphic fungi belonging to the family *Ophiostomataceae*. The most common etiological agent of sporotrichosis is *Sporothrix schenckii*, although other species, such as *Sporothrix brasiliensis*, *Sporothrix globosa*, and *Sporothrix mexicana*, have also been implicated in human infections [3,4].

Human sporotrichosis occurs in many parts of the world, with higher incidence rates reported in tropical and subtropical regions, including Latin America, Asia, and Africa. The disease typically affects individuals engaged in outdoor activities or occupations involving contact with soil, plants, or organic matter, such as agriculture, gardening, forestry, and floristry. Although commonly associated with traumatic inoculation from plant material (classic transmission), sporotrichosis can also be acquired through inhalation of conidia or via direct inoculation during contact with infected animals (zoonotic transmission) [1,5,6].

Brazil has been facing an epidemic of sporotrichosis in the last two decades, characterized by a dramatic increase in human, canine, and feline cases attributed to zoonotic transmission, especially in the metropolitan regions from the Southern and the South regions [1,7]. The epidemic has raised awareness among healthcare professionals, veterinarians, and public health authorities about the need for enhanced surveillance, diagnosis, and management of sporotrichosis [8,9].

Although human sporotrichosis has been reported in the states from Amazon, they were usually associated with classic transmission [10]. This article reports the rising incidence of zoonotic sporotrichosis in the Amazon State, Brazil.

## Methods

### Ethics statement

Ethical approval for this study was obtained from the Fundação Alfredo da Matta Ethics Committee (approval number 6.053.509).

An ecologic study was performed by assessing the official registers of human and animal sporotrichosis diagnosed in the state of Amazon from 2020 to 2023. In the state of Amazon, both human and animal sporotrichosis are mandatory reporting. For the official registry, the diagnosis of sporotrichosis is based on clinical and epidemiological aspects and/or laboratory findings.

Demographic data, habits, clinical, and fungal identification were examined from the human cases. Geographic data from human and animal cases were recorded. Both demographic and geographic information were collected from online Brazilian databases "Sistema de Informação de Agravos de Notificação" (SINAN) and "Fundação de Vigilância em Saúde do Amazonas" (FVS).

Cultures were performed by sowing clinical specimens from the active lesion (exudate and/or biopsy fragment) in test tubes containing Sabouraud agar with chloramphenicol and Mycosel agar. Culture plates were incubated at room temperature for 6 to 10 days. After satisfactory growth of the filamentous colony, morphological analysis was carried out to confirm the genus *Sporothrix*.

For *Sporothrix* speciation, real-time polymerase chain reaction technique (qPCR) was used. First, *Sporothrix* spp genomic DNA was extracted from the clinical specimen using a Qiagen Blood & Tissue purification kit; purified DNA was quantified and stored in a -20˚C freezer for later use in qPCR. For the detection and identification of species of *Sporothrix* (*S.brasilensis*, *S. schenckii* and *S. globosa*), TaqMan MGB-NFQ primers and probes were used. The following primers were used: CGTCTGAGCGTCTACTTCAACG and GGACGGCATCCATGGTACC. The probes used were CGATCGGCTTTGCTTTGGCCCTAGT (*S.brasilensis*), *TCCCAC CGTTTGGCAC* (*S. schenckii*) and CGTCACAGTTTTGGCACGATTCTAACAATTTTT (*S. globosa*).

Data were summarized according to the municipality of the origin, geolocalization of the cases, age, gender, occupation, contact with an infected animal, and topography of the lesions.

## Results

The annual records of 950 human cases and 2,823 animal cases of sporotrichosis from 2020 to 2023 are presented in Fig 1. The rise in human cases from 2021 to 2022 amounted to 304%, and from 2022 to 2023, it reached 249%. In comparison, the increase in animal cases was 428% and 345%, respectively, for the same periods (Fig 1).

Most of cases were from Manaus (96,9%) but also from other populous municipalities as Presidente Figueiredo (0,9%), Barcelos (0,8%), Iranduba (0,6%), Urucurituba (0,2%), Carauari (0,1%), Canutama (0,1%), Careiro da Várzea (0,1%), and Careiro (0,1%) (Fig 2).

Of the animals registered with sporotrichosis, 99.1% were cats (Fig 3).

Among the human cases, 60.9% were females, with a mean (SD) age of 38.9 (18.9) years. Most of the individuals were self-declared as brown (81.5%). Adults (19–59 y.o.) were the most affected group, accounting for 68.5% of the cases, followed by the elderly (>60 y.o.) with 15.4%, children (0–12 y.o.) with 8.8%, and teenagers (13–18 y.o.) with 7.3% of the cases. None of the individuals affected were engaged in occupations involving plant work, agriculture, gardening, or soil-related activities.

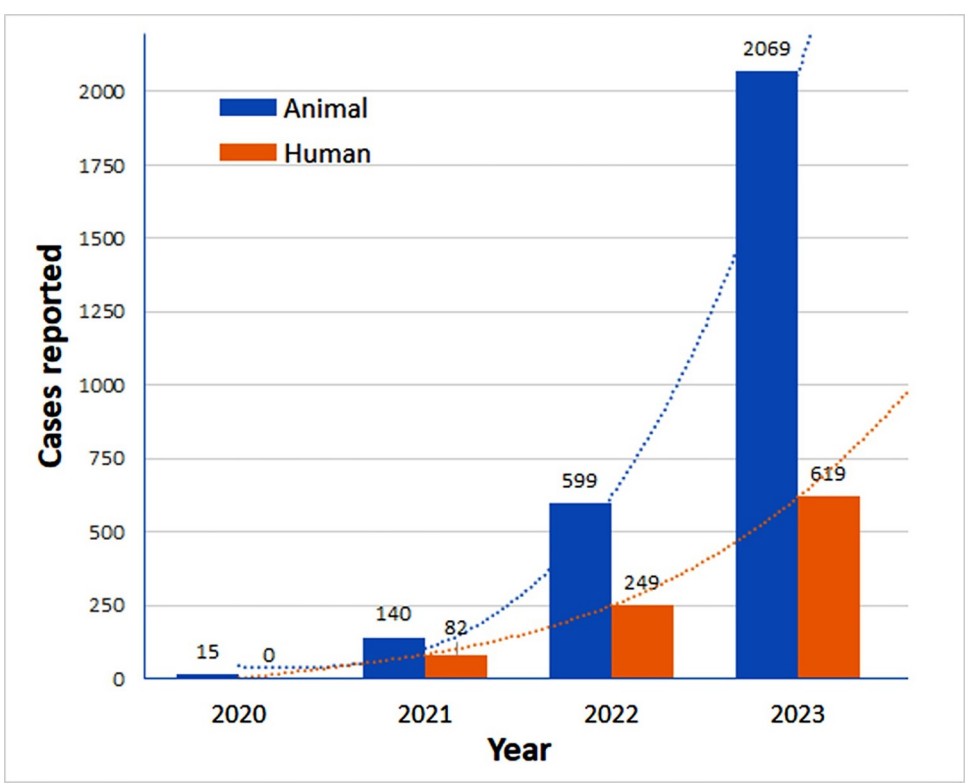

**Fig 1. Graphic curve representation of the exponential number of sporotrichosis cases in animals and humans, in Amazon State, Brazil, from 2021 to 2023 (Source: SINAN-NET).**

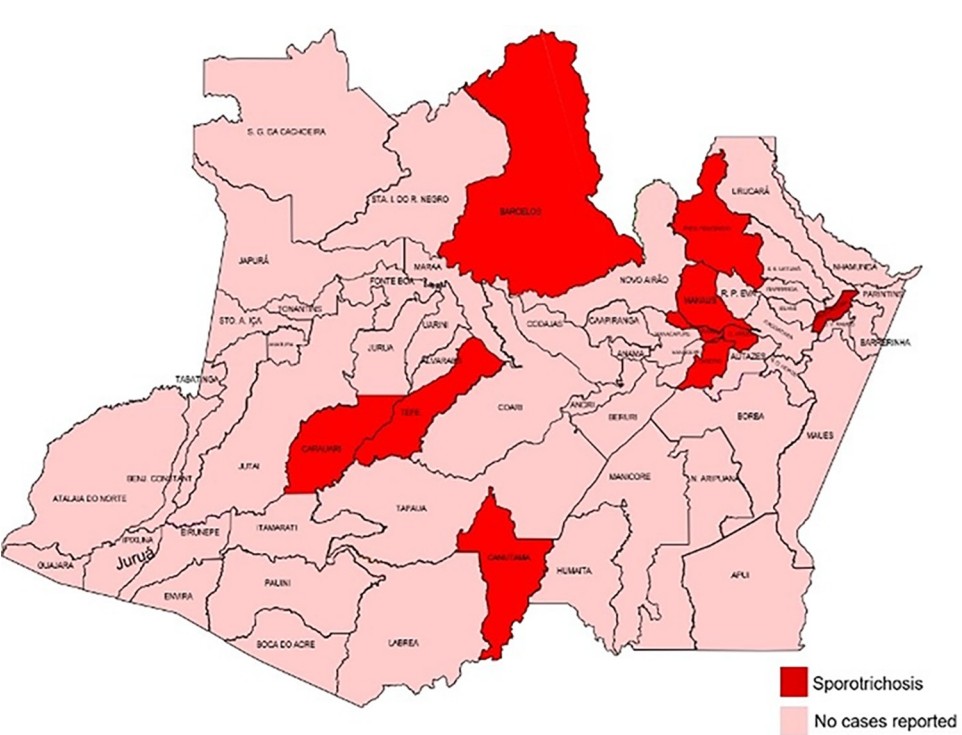

**Fig 2. Municipalities from the Brazilian Amazon Region with reported human sporotrichosis from 2021 to 2023 (Source: SINAN—NET) (Source of the basemap shapefile: IBGE https://www.ibge.gov.br/geociencias/ organizacao-do-territorio/malhas-territoriais/15774-malhas.html).**

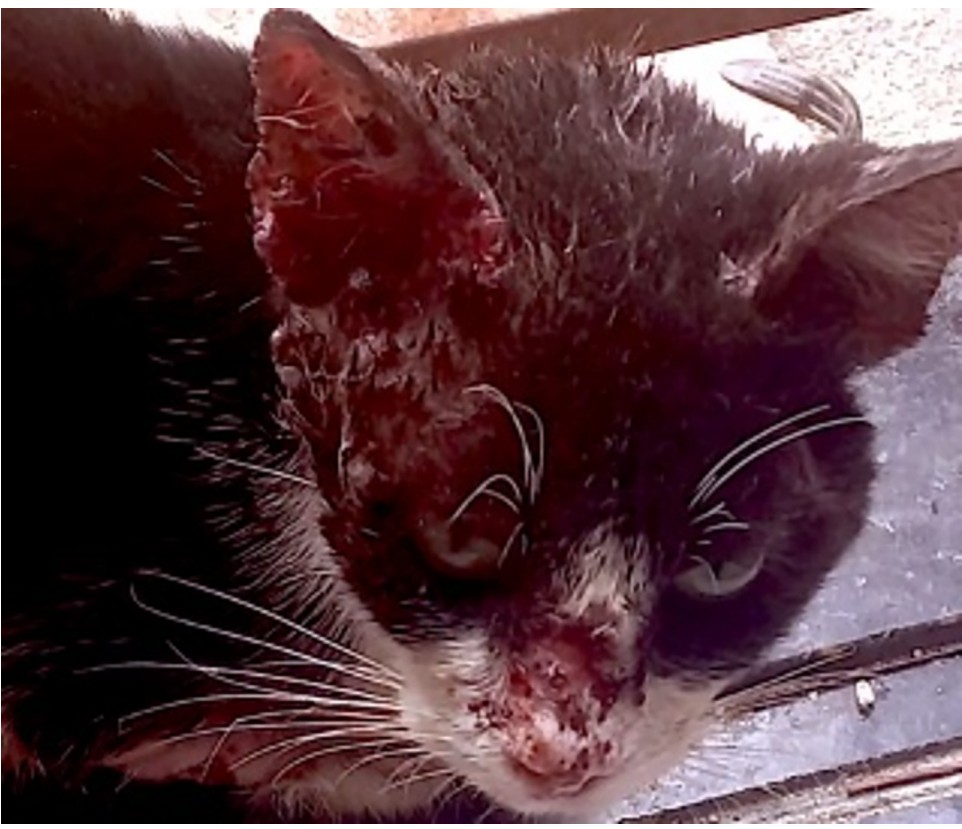

**Fig 3. Sporotrichosis in a cat from Manaus presenting with extensive facial ulceration involving the ears.** Note the emaciated appearance of the animal.

Regarding the most affected site, it was observed that the upper limbs (39.6%), followed by the hands (29.6%), lower limbs (26.3%), chest (4.8%), feet (3.7%), head (3.2%), and abdomen (1.8%) were reported (Fig 4).

Of cases where cutaneous lesions were described, 412 patients presented with ulcers, 82 with nodules, and 76 with papules.

The cases reported contact with infected animals in 77.7% of instances, with cats being the most commonly cited animals (73.5%), followed by dogs (2.7%), or both (1.0%).

Regarding possible infection sites, the urban area is noteworthy with 93.8% cases, and the home environment was the most common with 79.8% cases, followed by work with 4.4% and leisure places with 1.8% cases.

Only 66.7% of the individuals diagnosed with sporotrichosis collected samples for the mycologic exam. Of 78 cultures positive for *Sporothrix spp*, 65.4% were identified as *S. brasiliensis*.

All cases were treated with approved systemic antifungals for sporotrichosis therapy. Itraconazole is also the drug of choice for the treatment of infected cats. To date, no human death due to sporotrichosis has been registered in the state of Amazonas.

Up to May/2024, 428 confirmed cases of human and 704 animal cases of sporotrichosis were reported in the state of Amazonas.

## Discussion

Since the initial reports of zoonotic sporotrichosis from São Paulo and Rio de Janeiro in the late 1990s, numerous municipalities in the Southern and South regions, such as Belo

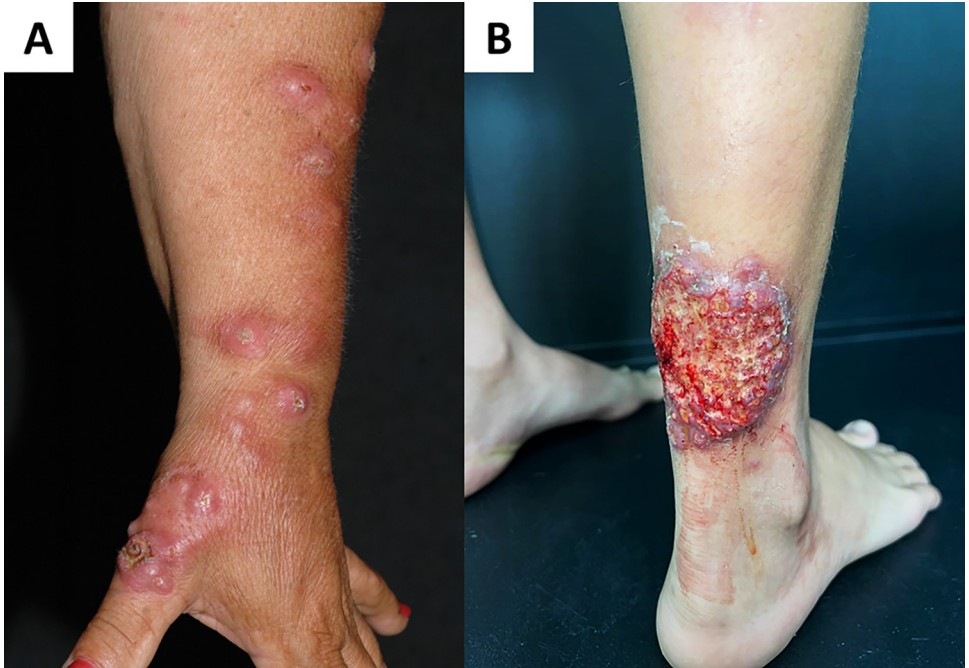

**Fig 4. Human sporotrichosis.** A. Lymphocutaneous sporotrichosis. Ulcer with infiltrate borders on the thumb (local of inoculation–cat bite), followed by ascending (lymphangitic) erythematous nodules. B. Leg ulcer in an adult who refers a previous scratch from an infected cat.

Horizonte, Curitiba, and Porto Alegre, have documented escalating series of urban transmission associated with infected felines. More recently, a series of zoonotic sporotrichosis were reported in Brasília (Central-West Region), as well as in municipalities across the Northeast Region, including Natal, Salvador, and Recife. These findings illustrate the progressive spread of the epidemic throughout major Brazilian metropolises [11–15].

Manaus, with a population exceeding 2 million inhabitants, stands as the largest city in the Amazon region, serving as a pivotal communication hub connecting Brazil's North Region with neighboring countries such as Peru, Colombia, Venezuela, and various Caribbean nations. The rising incidence of zoonotic sporotrichosis in Manaus mirrors the epidemic trajectory witnessed in other densely populated metropolises.

The Amazon, known for its vast rainforests and biodiversity, has experienced significant urbanization over the past few decades. This urban growth is driven by various factors, including economic development, rural-to-urban migration, infrastructure projects, and government policies. Despite the opportunities offered by urbanization, it also brings challenges such as environmental degradation, pressure on natural resources, social inequalities, and inadequate urban infrastructure. The unplanned expansion of cities and towns can lead to deforestation, habitat loss, pollution, and increased vulnerability to natural disasters and disease outbreaks [16–19]. The presence of large colonies of feral cats in urban and peri-urban areas has facilitated the dissemination of *Sporothrix* species, contributing to the epidemic spread of sporotrichosis among humans and animals.

Zoonotic transmission is most commonly associated with dogs and cats, particularly domestic cats, which serve as the primary reservoirs for the fungus. Cytologic exams and cultures taken from skin lesions, nasal cavities, oral cavities, and nails of domestic cats frequently yield positive results, supporting the hypothesis of transmission through scratches or bites

inflicted by infected animals. Additionally, contact with secretions containing infectious microorganisms further contributes to the spread of the disease [20].

Cat-transmitted sporotrichosis caused by *S. brasiliensis* has become a major public health concern and presents a distinct divergence from the traditional epidemiology of sporotrichosis [21]. As the presence of infected felines in densely populated urban areas precedes human infection, our data warns of an exponential rise in new cases of human and animal sporotrichosis in the Amazon region.

The pathogenic *Sporothrix* species share certain morphological characteristics but also exhibit differences that are important for the accurate understanding of their epidemiology and pathogenicity. Regarding their geographic distribution and ecological niches, *S. brasiliensis* is predominantly found in tropical and subtropical regions of South America, particularly in Brazil, where it is considered the main causative agent of the ongoing epidemic of sporotrichosis. On the other hand, *S. globosa* is more commonly associated with sporotrichosis in Asia. Genetic diversity and antifungal susceptibility also differ among the species. The cultures of *S. brasiliensis* exhibit slower growth compared to other species at 20˚C (soil temperature), but they grow more rapidly than others at >36˚C (mammalian tissue temperature), thereby favoring zoonotic transmission [4]. Concerning their virulence *S. brasiliensis* has been found to exhibit increased virulence compared to other species, leading to more severe clinical manifestations in infected individuals [22].

The upper extremities were the most frequently affected body site in the present study (69%), followed by the lower extremities (30%). Moreover, infection in urban areas (94%), contact with infected cats (78%), and the high identification of *S. brasiliensis* among positive cultures reinforce the expected pattern in a zoonotic epidemic, as observed in other Brazilian metropolises [15].

The epidemic of sporotrichosis in Brazil highlights the complex interplay between humans, animals, and the environment in the epidemiology of zoonotic fungal infections. Enhanced surveillance, diagnosis, and management of sporotrichosis are imperative to control the spread of the disease and minimize its impact on public health [21]. Nevertheless, as clinical manifestations in humans can vary from typically localized ulcers on the extremities to atypical immunoreactive forms, direct microscopic examination, and histopathology are insufficient for diagnosis, primary care services lack access to fungal culture, making a prompt diagnosis of sporotrichosis challenging in the public health system.

In conclusion, there has been an escalating incidence of animal and human sporotrichosis in the Brazilian Amazon state in the last four years, whose prevalent agent was *S. brasiliensis*. Collaborative efforts among healthcare professionals, veterinarians, and public health authorities are essential to implement effective control measures, educate at-risk populations, and promote responsible guidance for pet guardians to mitigate the burden of epidemic sporotrichosis in Brazil.

## Author Contributions

**Conceptualization:** Maria Zeli Moreira Frota, Helio Amante Miot, Carolina Talhari, Valderiza Lourenço Pedrosa.

**Data curation:** Viviany Araujo Mesquita, Sinesio Talhari, André Luiz Leturiondo, Guilherme Caldas de Souza, Débora Cristina de Lima Fernandes, Carolina Talhari.

**Formal analysis:** Sinesio Talhari, André Luiz Leturiondo, Guilherme Caldas de Souza, Euzenio Moreira de Brito, Suanni Lemos de Andrade, Débora Cristina de Lima Fernandes, Maria Zeli Moreira Frota, Rossilene Conceição da Silva Cruz, Juliana de Andrade Rebouças Guimarães, Helio Amante Miot, Carolina Talhari, Valderiza Lourenço Pedrosa.

**Funding acquisition:** Carolina Talhari.

**Investigation:** Viviany Araujo Mesquita, Sinesio Talhari, André Luiz Leturiondo, Guilherme Caldas de Souza, Euzenio Moreira de Brito, Suanni Lemos de Andrade, Débora Cristina de Lima Fernandes, Maria Zeli Moreira Frota, Rossilene Conceição da Silva Cruz, Juliana de Andrade Rebouças Guimarães, Helio Amante Miot, Carolina Talhari, Valderiza Lourenço Pedrosa.

**Methodology:** Viviany Araujo Mesquita, Sinesio Talhari, André Luiz Leturiondo, Guilherme Caldas de Souza, Euzenio Moreira de Brito, Suanni Lemos de Andrade, Débora Cristina de Lima Fernandes, Maria Zeli Moreira Frota, Rossilene Conceição da Silva Cruz, Juliana de Andrade Rebouças Guimarães, Helio Amante Miot, Carolina Talhari, Valderiza Lourenço Pedrosa.

**Project administration:** Carolina Talhari, Valderiza Lourenço Pedrosa.

**Resources:** Carolina Talhari.

**Supervision:** Helio Amante Miot, Carolina Talhari, Valderiza Lourenço Pedrosa.

**Validation:** Viviany Araujo Mesquita, Guilherme Caldas de Souza, Euzenio Moreira de Brito, Suanni Lemos de Andrade, Débora Cristina de Lima Fernandes, Maria Zeli Moreira Frota, Rossilene Conceição da Silva Cruz, Juliana de Andrade Rebouças Guimarães, Helio Amante Miot, Carolina Talhari, Valderiza Lourenço Pedrosa.

**Visualization:** Viviany Araujo Mesquita, Sinesio Talhari, Guilherme Caldas de Souza, Euzenio Moreira de Brito, Suanni Lemos de Andrade, Débora Cristina de Lima Fernandes, Maria Zeli Moreira Frota, Rossilene Conceição da Silva Cruz, Juliana de Andrade Rebouças Guimarães, Helio Amante Miot, Carolina Talhari, Valderiza Lourenço Pedrosa.

**Writing – original draft:** Viviany Araujo Mesquita, Sinesio Talhari, Helio Amante Miot, Carolina Talhari, Valderiza Lourenço Pedrosa.

**Writing – review & editing:** Viviany Araujo Mesquita, Carolina Talhari.

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
