## [Decision Letter · Decision Letter 0]

10 May 2024

Dear Dr. Talhari,

Thank you very much for submitting your manuscript "Zoonotic Sporotrichosis Outbreak: Emerging Public Health Threat in the Brazilian Amazon." for consideration at PLOS Neglected Tropical Diseases. As with all papers reviewed by the journal, your manuscript was reviewed by members of the editorial board and by several independent reviewers. The reviewers appreciated the attention to an important topic. Based on the reviews, we are likely to accept this manuscript for publication, providing that you modify the manuscript according to the review recommendations. 

Sincerely,

Joshua Nosanchuk, MD

Section Editor

Joshua Nosanchuk

Section Editor

Reviewer's Responses to Questions

**Key Review Criteria Required for Acceptance?**

**Methods**

-Are the objectives of the study clearly articulated with a clear testable hypothesis stated?

-Is the study design appropriate to address the stated objectives?

-Is the population clearly described and appropriate for the hypothesis being tested?

-Is the sample size sufficient to ensure adequate power to address the hypothesis being tested?

-Were correct statistical analysis used to support conclusions?

-Are there concerns about ethical or regulatory requirements being met?

Reviewer #1: The methods were clearly defined. The study design is adequated and address the objectives of the study. The statiscal analysis udes was adequated.

Reviewer #2: See Editorial and data presentation modifications

**Results**

-Does the analysis presented match the analysis plan?

-Are the results clearly and completely presented?

-Are the figures (Tables, Images) of sufficient quality for clarity?

Reviewer #1: The results were clearly presented including some nice figures. Please, insert a graphic curve representation of the exponetial number of the cases during the period of the study.

Reviewer #2: See Editorial and data presentation modifications

**Conclusions**

-Are the conclusions supported by the data presented?

-Are the limitations of analysis clearly described?

-Do the authors discuss how these data can be helpful to advance our understanding of the topic under study?

-Is public health relevance addressed?

Reviewer #1: The authors discussed the data very clearly, annd the public health was addressed very well.

Reviewer #2: See Editorial and data presentation modifications

**Editorial and Data Presentation Modifications?**

Reviewer #1: Please, insert a graphic curve representation of the exponetial number of the cases during the period of the study.

Reviewer #2: Please consider the following suggestions as a contribution to improve the article for the journal readers.

1. Although the title suggests that authors will show data from the Braziliian Amazon region, in Methods it seems that data came only from Amazon state. Is that true?

2. Authors should change animal owners and ownership for guardians since nobody is the owner of other living being.

3. Figure 1 in Introduction is out of purpose and unnecessary. It should be withdrawn. 

4. I did not see a statement about Ethics Committee. Could you please provide it?

5. Methods: “An ecologic study was performed by assessing the official registers of human and animal sporotrichosis diagnosed in the state of Amazon from 2020 to 2023.” Could you please provide which official registers and if it is available for everyone? Were they state or federal? SINAM? GAL? For more reliable data please specify better in the text.

6. Methods: “In the state of Amazon, both human and animal diseases are of mandatory

 reporting.” The authors mean all human and animal disease or this is just for sporotrichosis? Since when has this been done? Do you think it is underreported? Are all demographic and clinical data reported in the official registers? Please, add a comment.

7. How were the diagnosis made? Was it an epidemiological inference? 

There were 950 human cases diagnosed, but only 633 had culture made. Of these only 78 (12.3%) were positive for Sporothrix spp, and 65 out of 78 were identified as S. brasiliensis. Could you explain these data better? Which one was the species of these 13 Sporothrix not brasiliensis identified?

8. Statistical analysis were from the 950 patients?

9. Please standardize the binomial writing S. braSiliensis and not braZiliensis throughout the text.

10. Do official records report treatment outcome? Authors reported all cases were cured.

11. There was no hospitalization in these 950 patients? Anyone immunosupressed?

**Summary and General Comments**

Reviewer #1: No more comments.

Reviewer #2: To the Authors 

This is an important and well-written article with a special epidemiological interest about the spread of zoonotic sporotrichosis to Amazon in Brazil. However some issues should be regarded.

PLOS authors have the option to publish the peer review history of their article (what does this mean?). If published, this will include your full peer review and any attached files.

Reviewer #1: Yes: Paulo Ricardo Criado

Reviewer #2: No

Figure Files:

Data Requirements:

Reproducibility:

References

---

## [Editor Report · Decision Letter 1]

27 Jun 2024

Dear Dr. Talhari,

We are pleased to inform you that your manuscript 'Zoonotic Sporotrichosis Outbreak: Emerging Public Health Threat in the Amazon State, Brazil.' has been provisionally accepted for publication in PLOS Neglected Tropical Diseases.

Best regards,

Joshua Nosanchuk, MD

Section Editor

Editorial note:  The authors are to be commended for their thoughtful and professional responses to the comments on the prior review.

---

## [Editor Report · Acceptance letter]

15 Jul 2024

Dear Dr. Talhari,

We are delighted to inform you that your manuscript, "Zoonotic Sporotrichosis Outbreak: Emerging Public Health Threat in the Amazon State, Brazil.," has been formally accepted for publication in PLOS Neglected Tropical Diseases.

Best regards,

Shaden Kamhawi

co-Editor-in-Chief

Paul Brindley

co-Editor-in-Chief
